# Dietary marine hydrolysate alleviates D-galactose-induced brain aging by attenuating cognitive alterations, oxidative stress and inflammation through the AGE-RAGE axis

Camille Mougin[1,2], Mathilde Chataigner[1,2], Céline Lucas[3], Véronique Pallet[1], Elodie Bouvret[2], Corinne Joffre[1], Anne-Laure Dinel[1,3]*

1 Nutrineuro, UMR 1286, Université Bordeaux, INRAE, Bordeaux INP, Bordeaux, France, 2 Abyss Ingredients, Caudan, France, 3 NutriBrain Research and Technology Transfer, NutriNeuro, Bordeaux, France

* anne-laure.dinel@inrae.fr

**Data Availability Statement:** All relevant data are within the manuscript.

## Abstract

Aging represents a natural and unavoidable phenomenon in organisms. With the acceleration of population aging, investigations into aging have garnered widespread global interest. One of the most striking aspects of human aging is the decline in brain function, a phenomenon intricately tied to the onset of neurodegenerative conditions. This study aimed to assess the impact of a fish hydrolysate, rich in low-molecular-weight peptides and n-3 LC-PUFAs, on cognitive function, inflammatory response, and oxidative stress *via* the AGE-RAGE axis in a mouse model of accelerated aging. This model induces cognitive decline and biochemical alterations akin to those observed during natural aging. The findings revealed that fish hydrolysate exhibited a protective effect against cognitive impairment induced by D-galactose. This effect was associated with increased protein expression of SOD1 and decreased genetic expression of IL-6 and advanced glycation end products (AGE). Consequently, within the realm of preventive and personalized nutrition, fish hydrolysate emerges as a promising avenue for mitigating age-related declines in memory function.

## Introduction

Over the past decades, the rapid aging of the population in developed countries has drawn considerable attention, leading to notable adjustments in public health policies [1]. This aging process is often accompanied by cognitive decline, characterized by significant yet non-pathological alterations in memory, learning, and spatial recognition [2, 3]. These age-related cognitive changes can lead to degenerative brain disorders such as Alzheimer's disease and other forms of dementia, which can hasten the loss of independence in older adults [4, 5].

Aging is a complex process influenced by multiple factors, leading to a gradual deterioration in physiological function. Age-related cognitive alterations have been linked to chronic low-grade inflammation [6, 7]. Microglia, the brain's immune cells, become more sensitive to

**Funding:** This work is part of the Optimyss project, which has been funded by the National Agency of Research (ANR France) and Abyss Ingredients in the context of the national French project "Plan de relance" (ANR 21-PRRD-0058-01).The funders had no role in study design, data collection and analysis, decision to publish, or preparation of the manuscript.

**Competing interests:** Céline Lucas, Véronique Pallet, Corinne Joffre and Anne-Laure Dinel report no disclosures. Abyss Ingredients funds Camille Mougin, Mathilde Chataigner, and Elodie Bouvret. Mathilde Chataigner: employee of Abyss Ingredients; Elodie Bouvret: employee of Abyss Ingredients. This work is part of a collaborative project named Optimyss which has been funded by the National Agency of Research (ANR France) and Abyss Ingredients in the context of national French project "Plan de relance". Mathilde Chataigner and Elodie Bouvret work for Abyss Ingredients and provide the fish hydrolysate, described in the patent number FR3099339(B1) in which Mathilde Chataigner is cited as an inventor; Camile Mougin was recruited for this specific research program by NutriNeuro. This does not alter our adherence to PLOS ONE policies on sharing data and materials.

inflammation as individuals age, leading to heightened reactivity and prolonged production of pro-inflammatory cytokines such as interleukin-6 (IL-6), interleukin-1β (IL-1β) and tumor necrosis factor-alpha (TNF-α) [8, 9]. Increased levels of these cytokines can initiate a series of neuroinflammatory processes, including neuronal loss [10]. Additionally, there is extensive documented evidence indicating that oxidative stress and reactive oxygen species (ROS) are widely implicated in aging and associated neurodegenerative disorders [11–13]. Excessive ROS production or reduced antioxidant levels can lead to lipid peroxidation, DNA damage in the nucleus and mitochondria, and protein oxidation, resulting in significant cognitive impairment [14–17]. Finally, Monnier and Cerami introduced a Maillard theory of aging, suggesting that the gradual accumulation of advanced glycation end products (AGEs), could be a driving force behind aging [18–22]. They also proposed that this prolonged accumulation of AGEs might lead to alterations in protein structure and function, influencing various aging hallmarks [23, 24]. AGEs have also been known to interact with receptors for AGEs (RAGEs) resulting in the dysregulation of cell signaling, inflammatory response and oxidative response associated with neurodegenerative diseases of aging [25]. Considering these potential mechanistic understandings, substances with antioxidant and/or anti-inflammatory properties may offer promising avenues for preventing cognitive decline during aging.

Nutrition holds potential in addressing age-related cognitive decline. Nutrients such as n-3 long-chain polyunsaturated fatty acids (LC-PUFAs), specifically eicosapentaenoic acid (EPA) and docosahexaenoic acid (DHA), have been associated with enhanced cognitive function in older adults [26, 27]. Pre-clinical studies reinforce these findings, demonstrating that n-3 LC-PUFAs improve spatial learning and memory in aged mice [28–30]. The immunomodulatory properties of n-3 LC-PUFAs may underlie this benefit, as evidenced by reduced cytokine production in older individuals with higher blood pressure levels [28, 31]. Additionally, EPA functions as a peroxisome proliferator-activated receptor alpha (PPARa) agonist, enabling the modulation of several signaling pathways. This modulation includes the reduction of neuroinflammation and oxidative stress, both directly contributing to neuroprotection [32]. Moreover, low molecular weight peptides (< 1000 Da) derived from protein hydrolysates offer potential anti-inflammatory properties. Orally administered milk peptides suppress inflammatory factors in the hippocampus of mice with Alzheimer's disease [33]. Peptides sourced from soy, milk, salmon, and lupine have similarly demonstrated anti-inflammatory effects in mouse intestines and the abdominal aorta [34, 35]. Combining such peptides with n-3 LC-PUFAs from fish hydrolysate potentially alleviates age-related impairments. Indeed, our previous study has shown that supplementation with a fish hydrolysate rich in low-molecular-weight peptides and n-3 LC-PUFAs improved memory and social behavior affected by aging, likely through modulation of gut microbiota and corticosterone levels [36–38]. However, the mechanisms of action remain unclear.

In this context, our study aims to pursue the evaluation of the neuroprotective effects of fish hydrolysate supplementation, containing mainly low-molecular-weight peptides and n-3 LC-PUFAs, on cognitive function, inflammatory and oxidative response through the AGE-R-AGE axis in a mouse model of accelerated aging that induces cognitive decline and biochemical modifications similar to those altered during aging [39, 40].

## Materials and methods

### Animals

The study was performed on 11-week-old male C57Bl/6J mice obtained from Janvier Labs (Le Genest-Saint-Isle, France). The animals were reared in a standard 12-hour light/12-hour dark cycle, housed on cellulose litter in a controlled environment (21–23˚C, 40% humidity). They

had unrestricted access to food and water. All animal care and experimental protocols adhered to the EU Directive 2010/63/EU for animal experiments and received approval from the national ethical committee for animal care and use (approval ID A27756).

## Treatments and diets

Mice were randomly divided into three groups: saline with control diet ($n$ = 16), D-gal-treated with control diet ($n$ = 8), and D-gal-treated with diet enriched with fish hydrolysate, comprising 0.21% of the hydrolysate ($n$ = 16) (Table 1). Diets were provided by INRAE (Jouy en Josas, France). The diets were initiated when the mice were 11-week-old and were maintained throughout the entire 8-week experiment. D-galactose (165 mg/kg, dissolved in sterile 0.9% saline) was injected subcutaneously (s.c.) daily into mice for 8 weeks. All controls were given vehicle (0.9% saline). Consequently, by the end of the experiment, the mice were 19-week-old.

The fish hydrolysate, obtained from Abyss Ingredients (Caudan, France) originated from marine by-products and predominantly comprised low-molecular-weight peptides (<1000 Da) in addition to n-3 LC-PUFAs, such as DHA and EPA. The specific composition of the fish hydrolysate is detailed in patent number FR3099339(B1). The dosage of low-molecular-weight peptides (2.9 mg/mouse/day) was calculated based on the quantity of peptides (<1000 Da) typically provided by 1 g of hydrolysate in humans, as determined by our previous findings [38].

## Behavioral test

**Spatial recognition short-term memory in the Y-Maze.** Six weeks following the initiation of supplementation, spatial recognition memory was assessed using the Y-maze test, as outlined by Dellu et al. [41]. The apparatus comprised a Y-shaped maze with three arms (35 cm long and 10 cm deep), illuminated at 15 lx. External visual cues were positioned on the maze walls to enable mice to orient themselves in space. In the initial trial, one arm of the Y-maze was closed, allowing mice to explore the remaining two arms for 5 minutes. Following a 1-hour inter-trial interval (ITI), mice were reintroduced to the start arm for the second trial, during which they could explore all three arms for another 5 minutes. Starting arms and closed arms were randomly assigned for each mouse. Video tracking (SMART system; Bioseb, Vitrolles, France) was employed to analyze the time spent in different arms. Additionally, a recognition index was calculated to assess the animals' performance relative to chance (33%).

**Spatial learning and reference memory in the Morris water maze.** Spatial learning and memory were evaluated through the Morris water maze (MWM), following the procedures

**Table 1. Composition of the control and hydrolysate-enriched diets.**

| Components | Percent (%) | |
| --- | --- | --- |
| | Control Diet | Hydrolysate-enriched diet |
| Hydrochloric casein | 18 | 18 |
| Corn starch | 45.9 | 45.69 |
| Sucrose | 24 | 24 |
| Cellulose | 2 | 2 |
| Peanut Oil | 5 | 5 |
| Mineral Mix | 4 | 4 |
| Vitamin Mix | 1 | 1 |
| + DL methionine | 0.1 | 0.1 |
| + Vitamin A 5 UI/g | 5 UI/g | 5 UI/g |
| Hydrolysate | 0 | 0.21 |

outlined in Bensalem et al. [42] and Morris [43]. Initially, two familiarization days were con-
ducted. Mice were required to locate a visible platform in a small pool (60 cm diameter) sur-
rounded by white curtains, facilitating acclimatization to water and swimming (3 consecutive
trials per day; 60 s cut-off). Subsequently, visuomotor deficits were assessed during a day of
cued learning in the Morris water maze, where mice had to find a visible platform indicated by
a cue (6 trials per day; 90 s cut-off). During spatial learning, mice underwent training over four
consecutive days to learn the location of a submerged platform using distal extramaze cues (6
trials per day; 90 s cut-off). Imetronic videotracking system (Pessac, France) recorded the
latency, distance traveled to reach the platform, and the swim path for each trial. Spatial mem-
ory was then evaluated 24 hours after the last training session through a 60-second probe test
in the maze without the platform.

The SMART system (Bioseb, Vitrolles, France) was employed to quantify the distance tra-
versed in each of the four quadrants, with the quadrant containing the originally positioned
platform during spatial learning identified as the target quadrant. Furthermore, a recognition
index was calculated to assess the animals' performance relative to the anticipated chance level
(25%).

## Analysis of navigation strategies in the Morris water maze

For each trial in the spatial learning test, we categorized the navigation path following the sys-
tem detailed by Bensalem et al. Strategies were divided into two main types: nonspatial
(encompassing "global search" methods like "peripheral looping" and "random", "circling," as
well as "local search" approaches such as "scanning", "chaining", "repeated incorrect" and "focal
incorrect") and spatial (including "repeated correct", "focal correct", "spatial indirect" and "spa-
tial direct").

## Tissue processing

The mice received an analgesic (buprenorphine, 300 mg/kg) and were euthanized by an injec-
tion of pentobarbital (exagon, 300 mg/kg). After a transcardiac perfusion with phosphate-buff-
ered saline (PBS), hippocampus was collected and frozen at −80˚C until further analysis.

## Biochemical measurements

**Quantitative real-time PCR.**   The expression level of various target genes was assessed on
hippocampus using real-time quantitative PCR, following the protocol outlined by Rey et al.
[44]. These analyses were conducted on the hippocampus. In brief, total RNAs were extracted
from the hippocampus using TRIzol (Invitrogen, Life Technologies, Saint Aubin, France), and
the purity and quantity of RNA were measured for each sample using spectrophotometry
(Nanodrop, Life Technologies, Saint Aubin, France). One or two micrograms of RNA were
reverse transcribed into complementary DNA (cDNA) using Superscript IV (Invitrogen,
Life Technologies, Saint Aubin, France). Specific TaqMan Ⓡ primers were employed to
amplify the genes of interest, focusing on IL-6 (Mm00446190_m1), IL-1β (Mm00434228_m1),
TNF-α (Mm00443258_m1), CD11b (Mm00434455_m1), p53 (Mm01731290_g1),
p21 (Mm00432448_m1), p16 (Mm00494449_m1), p19 (Mm00486943_m1), RAGE
(Mm00545815_m1) and Gpx (Mm00656767_g1) for the hippocampus, with β-Actin
(Mm02619580_g1) as the housekeeping gene. Fluorescence readings were recorded on an ABI
PRISM 7500-sequence detection system (Applied Biosystems, Villebon sur Yvette, France).
Data were analyzed using the comparative threshold cycle (Ct) method, and the results were
expressed as relative fold change to control target mRNA expression, following the approach
outlined by Madore et al. [45].

**Western blot.** Hippocampal protein expression was assessed using the extraction protocol outlined by Simões et al. [46]. Protein concentration was determined through the bicinchoninic acid protein assay (Interchim, Montlucon, France) following the provided protocol. For analysis, proteins were separated on a 12% sodium dodecyl sulfate-polyacrylamide gel and subsequently transferred to nitrocellulose membranes. These membranes were then subjected to incubation with various primary antibodies: anti-AGE (ab23722, rabbit, Abcam, Cambridge, UK), anti-SOD1 (37385, rabbit, Cell Signaling, Leiden, The Netherlands), and anti-β-Actin as the housekeeping protein (4970, rabbit, Cell Signaling, Leiden, The Netherlands). The primary antibodies were detected using appropriate donkey horseradish peroxidase-conjugated secondary antibodies (711-035-152, Jackson Immunoresearch, Westgrove, PA, USA). The membranes were further incubated with a peroxidase revealing solution (SuperSignal West Dura, ThermoFisher, Waltham, MA, USA) and visualized using ChemiDoc MP (Biorad, Hercules, CA, USA). The protein levels of interest were normalized to β-Actin, and the results are expressed as relative expression.

## Statistical analysis

Statistical analyses were conducted with GraphPad Prism 7 (GraphPadSotfware, San Diego, CA, USA). Heatmap was performed using the web interface of MetaboAnalyst (MetaboAnalyst 5.0, https://www.metaboanalyst.ca/MetaboAnalyst/home.xhtml (accessed on 6 February 2024), Canada).

A one-sample t-test was used to compare the three experimental groups against the expected chance level of 33% in the Y-maze and against the chance level of 25% in Probe test of MWM.

A 1-way ANOVA with diet as the factor was assessed to analyze the cued learning in the MWM and the gene expression. Fisher's LSD post hoc test was applied when appropriate. A 2-way ANOVA with repeated measures was used to analyze the swim speed and spatial learning across training days in the MWM (factors: diet and days of learning). Subsequently, Fisher's LSD post hoc test was conducted.

All data were presented as means ± standard error of mean (SEM). Differences were considered significant when the p-value was $\leq 0.05$. Data are available on the following link https://doi.org/10.57745/WRKTLL.

## Results

### Fish hydrolysate supplementation prevents short-term and long-term spatial memory deficits and facilitates the adoption of spatial strategies in the process of spatial learning in D-gal mice

The impact of the supplement on short-term spatial memory was evaluated through a Y-maze test with a 1-hour ITI. The recognition index of the saline group exceeded the chance level (33%), indicating an absence of spatial memory alterations ($p$ = 0.05) (Fig 1A). In contrast, the recognition index of the D-gal group was not different from the chance level, suggesting cognitive alteration. Interestingly, the FH group had a recognition index significantly higher than chance level (33%)($p$ = 0.04). Consequently, FH supplementation prevented the cognitive deficit induced by D-gal injection.

Subsequently, we examined the influence of the supplementation on spatial learning and long-term memory using the MWM. Given a higher swimming speed in the FH group compared to the adult control group (diet effect, $p$ = 0.03), we chose to measure the distance traveled to reach the platform as a more suitable indicator of spatial learning acquisition. All

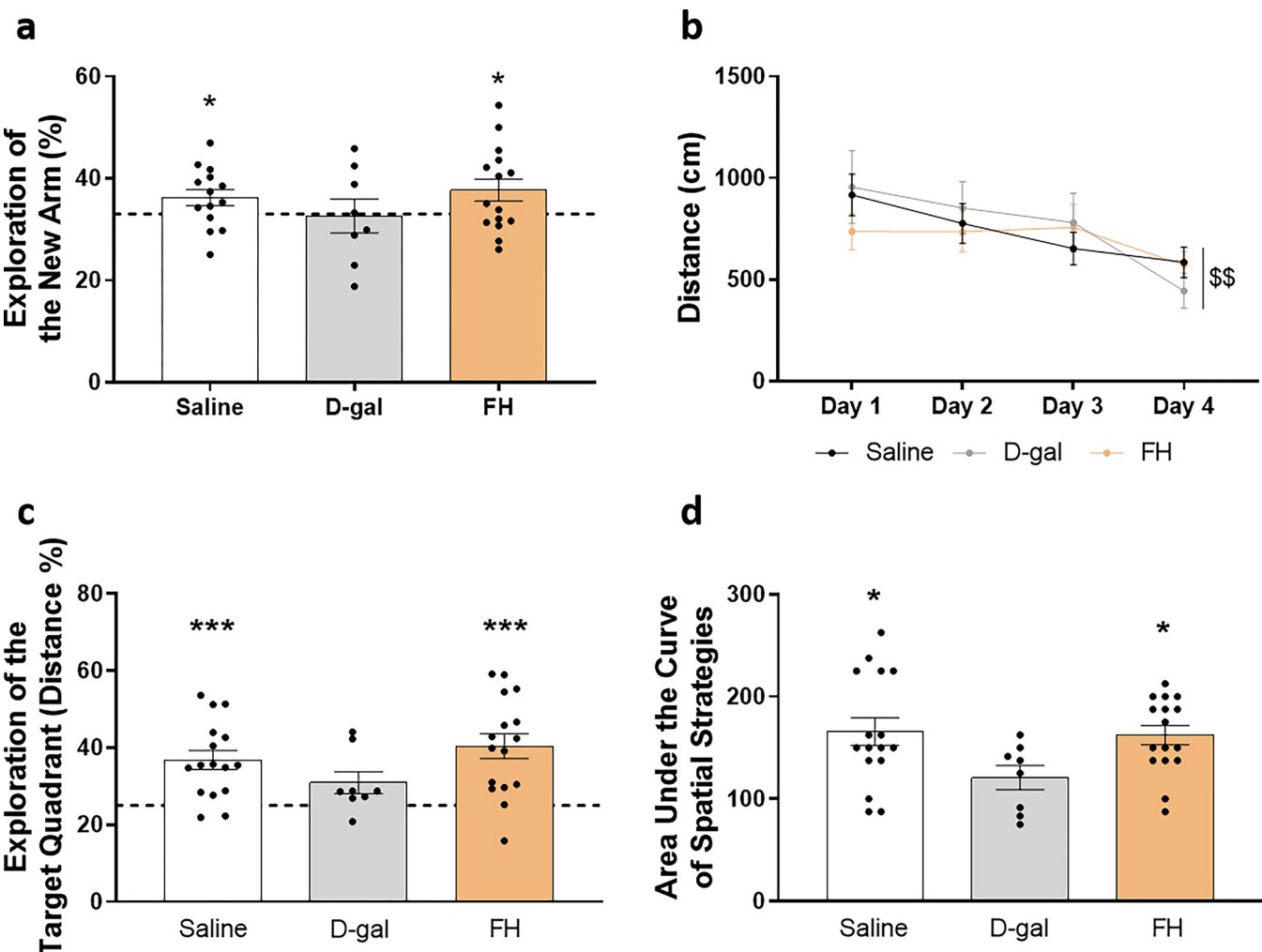

**Fig 1. Fish hydrolysate supplementation prevents short-term and long-term spatial memory deficits and promotes the use of spatial strategies during spatial learning in D-gal mice.** (a) Recognition index of the new arm after a 1h ITI in saline group, D-gal group and FH group. The dotted line corresponds to chance level (33%) (* $p \leq 0.05$ vs. chance level by one sample t-test). (b) Distance covered to reach the platform over the 4 consecutive days of spatial learning (day effect $ $ $ p < 0.01$ by 2-way ANOVA with repeated measures). (c) Percentage of distance travelled in quadrants during the probe test. The dotted line represents chance level (25%) *** $p < 0.001$ vs. chance level by one sample t-test. (d) Area under the curve of spatial strategies. n = 8–16 per group.

groups covered similar distances to reach the platform, indicating comparable visual abilities. Moreover, there was a significant reduction in the distance covered by all groups over the four days of training (day effect, $p < 0.01$), indicating a successful learning of the platform location (Fig 1B). Spatial memory was evaluated 24 hours after the final day of spatial learning. Related to the chance level of 25%, the saline group covered more distance in the target quadrant than in the others ($p < 0.001$), while the D-gal group did not show a significant difference ($p > 0.05$) (Fig 1C). These findings suggested a memory impairment in the D-gal group, as the mice failed to recognize the target quadrant. Notably, FH supplementation effectively prevented this issue since the supplemented mice covered more distance in the target quadrant compared to the chance level of 25% ($p < 0.001$).

Analysis of the navigation patterns revealed a shift in strategies from non-spatial to spatial over the learning days for all mice in the four groups. The area under the curve (AUC) of spatial strategies was significantly decreased by D-gal treatment ($p = 0.03$), and this was rescued by FH supplementation ($p = 0.04$) (Fig 1D).

## Fish hydrolysate supplementation modulates gene expression in D-gal mice

To understand the mechanisms underlying the prevention of memory deficits observed in the supplemented group, we analyzed the expressions of 12 genes in the hippocampus. These genes play roles in various pathways, including inflammation, senescence, AGEs, and antioxidant defenses.

The expression of IL-6, IL-1β, TNF-α and CD11b genes involved in the inflammatory response was analyzed. IL-6 expression was upregulated by D-gal treatment ($p = 0.05$) (Fig 2A). Interestingly, the expression of this gene was significantly down-regulated in the FH group compared to the D-gal group ($p = 0.01$) and restored to a similar expression level as in the saline group. The expression of IL-1β, TNF-α and CD11b was not changed by D-gal treatment and FH supplementation.

The senescent gene expression p16, p21, p19 and p53 was not modified by D-gal treatment and FH supplementation (Fig 2B).

The expression of AGE implicated in glycation was upregulated by D-gal treatment compared to the saline group ($p = 0.004$) and restored by FH supplementation ($p = 0.03$) while the expression of their receptors RAGE was not changed by D-gal treatment or FH supplementation (Fig 2C).

Although D-gal treatment did not affect the expression of Gpx and SOD1 related to antioxidant defenses, FH supplementation increased SOD1 protein expression compared to the D-gal group ($p = 0.03$) (Fig 2D).

To better understand the link between those pathways, a multivariate analysis was conducted. The outcome, represented as a heatmap, illustrated the result of hierarchical clustering (Fig 3A). Row dendrograms depicted the distance or similarity among gene expression and behavioral results. Column dendrograms illustrated the distance or similarity between groups and their respective clustering calculation nodes. Row dendrograms distinguished two clusters of parameters. The first cluster comprised downregulated variables (highlighted in yellow) in the D-gal group compared to the saline group. This cluster included protein related to antioxidant defenses (SOD1), as well as behavioral variables that were altered in D-gal mice (MWM, Y-maze, and AUC of spatial strategies). The second cluster comprised upregulated parameters (highlighted in purple) in the D-gal group compared to the saline group, notably including IL-6 and AGE. Subsequently, column dendrograms delineated distinct clusters for the saline group and the D-gal group; and affirmed the similarity between the saline group and FH group. Notably, FH supplementation reinstated performance levels in MWM, Y-maze, and the AUC of spatial strategy parameters to those observed in the saline group. Furthermore, it restored parameters associated with the inflammatory response (IL-6), antioxidant defenses (SOD1), and advanced glycation end-products. Based on these findings, the FH group and the saline group were clustered together. Further classification revealed that the D-gal group did not align with any other group, indicating a greater disparity from the other two groups.

Principal component analysis (PCA) confirmed the heatmap results (Fig 3B). Component 2 was negatively correlated with behavioral tests (Y-Maze, r = -0.21 and MWM, r = -0.18) and SOD1 (r = -0.24) (Fig 3C). This component was also positively correlated with AGE and their receptor (AGE, r = 0.49 and RAGE, r = 0.36). Additionally, on component 2, the saline group and the FH group had identical profiles. Both groups were positioned on the side of good behavioral results, high protein expression of SOD1 and low gene expression of AGE and RAGE. The D-gal group was positioned in the opposite way. Component 1 was negatively correlated with inflammation parameters (IL-6, r = -0.41; IL-1β, r = -0.39; TNF-α, r = -0.42 and CD11b, r = -0.38). On this component 1, the saline group and the FH group showed different profiles. The FH group was positioned on the side of a stronger inflammatory response unlike the saline and D-gal groups.

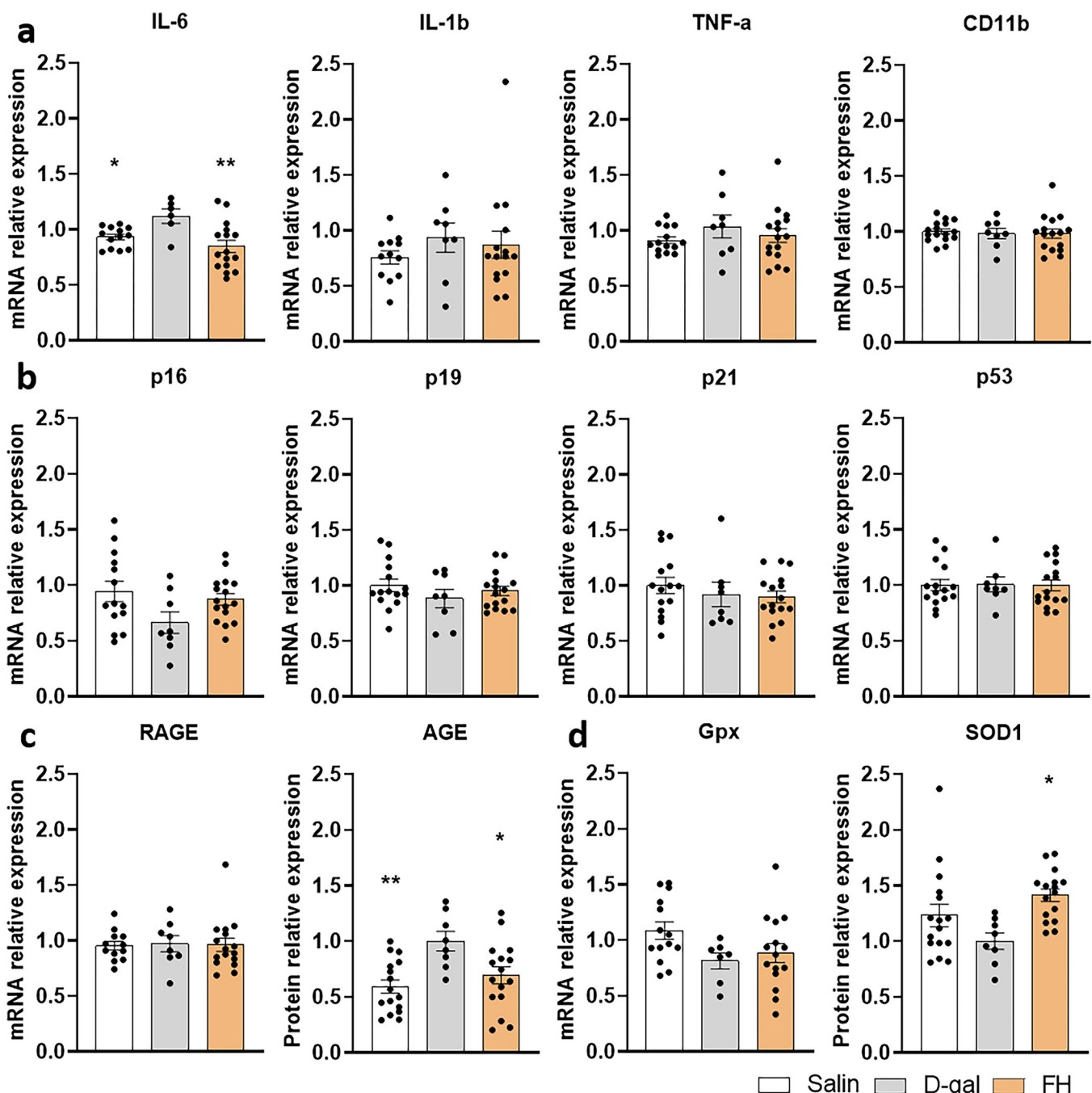

**Fig 2. Fish hydrolysate supplementation modulates gene and protein expression in D-gal mice.** (a) Inflammatory pathway. (b) Senescence pathway. (c) Advanced glycation end-product pathway. (d) Antioxidant defense pathway. * $p \leq 0.05$, ** $p < 0.01$ vs. D-gal. n = 8–16 per group.

## Discussion

Aging is a critical biological phenomenon with substantial medical and societal implications. It represents a significant challenge to develop innovative strategies for healthy aging. These strategies aim to reverse age-related changes once they occur or to mitigate and prevent aging-related alterations [47]. Recently, AGEs have been implicated in neurodegeneration and

**a**

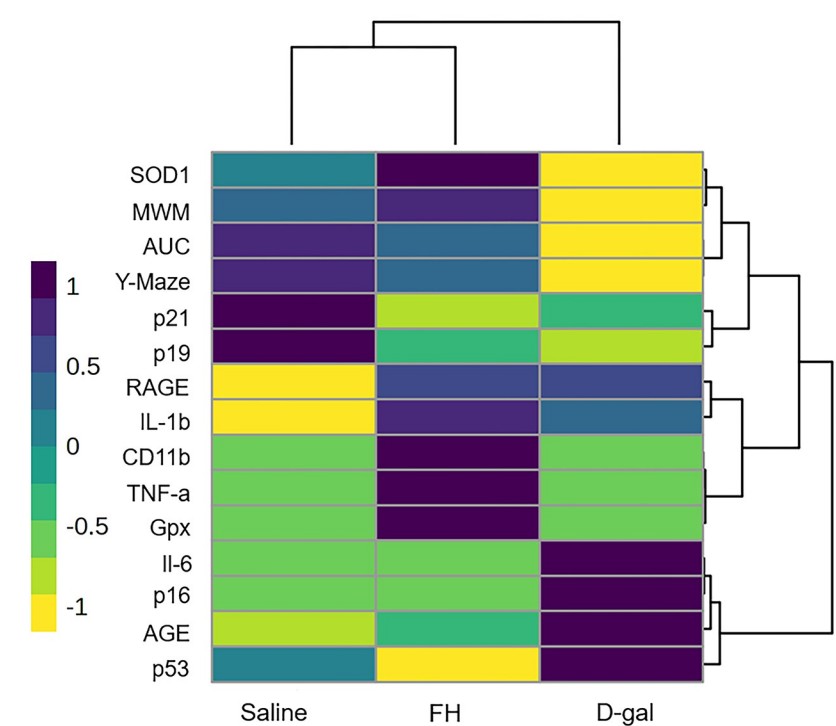

**b**                                          **c**

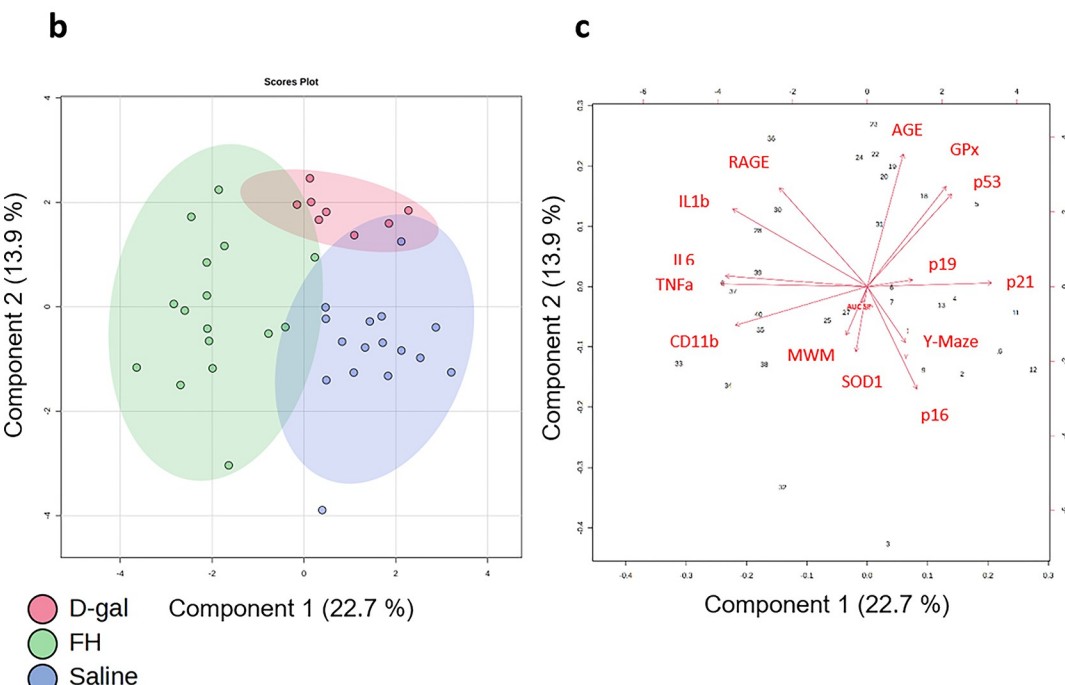

**Fig 3. Multivariate analysis.** (a) Heatmap analysis combined with hierarchical clustering analysis. Gene expression was determined by RT-qPCR. Each row represents one gene and each column represents one group. Purple indicates higher expression and yellow indicates lower expression. The global gene expression profile was compared after the end of 8-week supplementation, while D-gal group was used for the baseline gene expression. (b) Principal Component Analysis (PCA). (c) Variables factor map.

associated with neurodegenerative diseases. AGEs have also been known to interact with their specific receptor (RAGE) resulting in the dysregulation of cell signaling, inflammatory response and oxidative response [48]. Nutrients with antioxidant and/or anti-inflammatory properties emerge as a promising approach to combat cognitive decline in aging [49, 50]. In this study, our aim was to investigate the effect of a fish hydrolysate containing low-molecular-weight peptides and n-3 LC-PUFAs using a D-gal mice model to assess its impact on cognition, inflammatory and oxidative response through the AGE-RAGE axis. We demonstrated that supplementation with FH effectively prevented cognitive impairments caused by chronic D-gal injection (Fig 4). FH supplementation restored the markers evaluated to levels similar to those of control animals. Additionally, biochemical analyses revealed that FH supplementation induced reductions in D-gal induced AGE and IL-6 genetic expression, along with an increase in SOD1 protein expression in the brain. These results highlighted the beneficial effects of FH supplementation on cognitive functions by regulating oxidative stress and inflammation pathways through the AGE-RAGE axis.

The D-gal–induced aging mouse model has been a commonly used model of accelerated aging for studying the mechanisms of brain aging and anti-aging therapeutics in animal studies [39, 40]. Consistent with the literature, D-gal treatment induced short-term and long-term memory deficits. Indeed, the chronic injection of D-gal for 6–10 weeks in rodents induces a progressive decline in learning and memory abilities [51–55]. It also induces a significant increase in latency to reach the platform and a decrease in swim speed in mice models during the MWM test and a reduction of the percentage of novel arm entries during the Y-Maze [55, 56]. This model of accelerated aging induces memory alterations comparable to those recorded in an aged mouse model [38]. In our study, PCA showed that recognition indices of behavioral tests were negatively correlated with D-gal profile and positively correlated with saline and FH profiles. Indeed, FH supplementation prevented these deficits by increasing recognition indices in the MWM and Y-Maze. Our results can be compared with other interventional studies questioning cognitive decline. In human, a meta-analysis demonstrates that physical exercise (aerobic and resistance exercise of at least moderate intensity), improves cognitive function in the over 50s, regardless of the cognitive status of participants [57]. Moreover, several publications from our lab demonstrate the impact of dietary supplements on cognitive decline in mice and human [28, 36, 58, 59].

Furthermore, understanding the molecular mechanisms involved in memory deficits is important to define the efficacy of FH supplementation to prevent age-related diseases. When D-gal accumulates in the body, it undergoes oxidation by galactose oxidase, forming aldehydes and hydrogen peroxide ($H_2O_2$). However, an excessive supply of D-gal disrupts this metabolism [53, 60]. Indeed, D-gal, as a reducing sugar, readily reacts with the free amines of amino acids in proteins and peptides, forming AGEs through non-enzymatic glycation both *in vivo* and *in vitro* [61]. In our study, we showed that FH supplementation reduced the expression of AGEs induced by D-gal administration. Evidence has demonstrated that AGEs increase with age and have been implicated in the neuropathological lesions observed in some age-related conditions such as Alzheimer's disease [60]. Furthermore, histochemical analysis of Parkinson's patients indicates elevated levels of AGEs and RAGE in the frontal cortex compared to controls [62]. Moreover, PCA showed that AGEs level was negatively correlated with cognitive performance. Research has indicated a direct link between peripheral AGE levels and cognitive decline in older individuals [63]. Thus, by negatively regulating AGEs, FH supplementation specifically contributed to the prevention of cognitive disorders.

As hippocampus is one of the main structures involved in cognition (consolidation of episodic memory and context-dependent spatial learning) and as it is strongly impaired during aging (impairment including inflammation and oxidation) [28, 64–66], we measured the

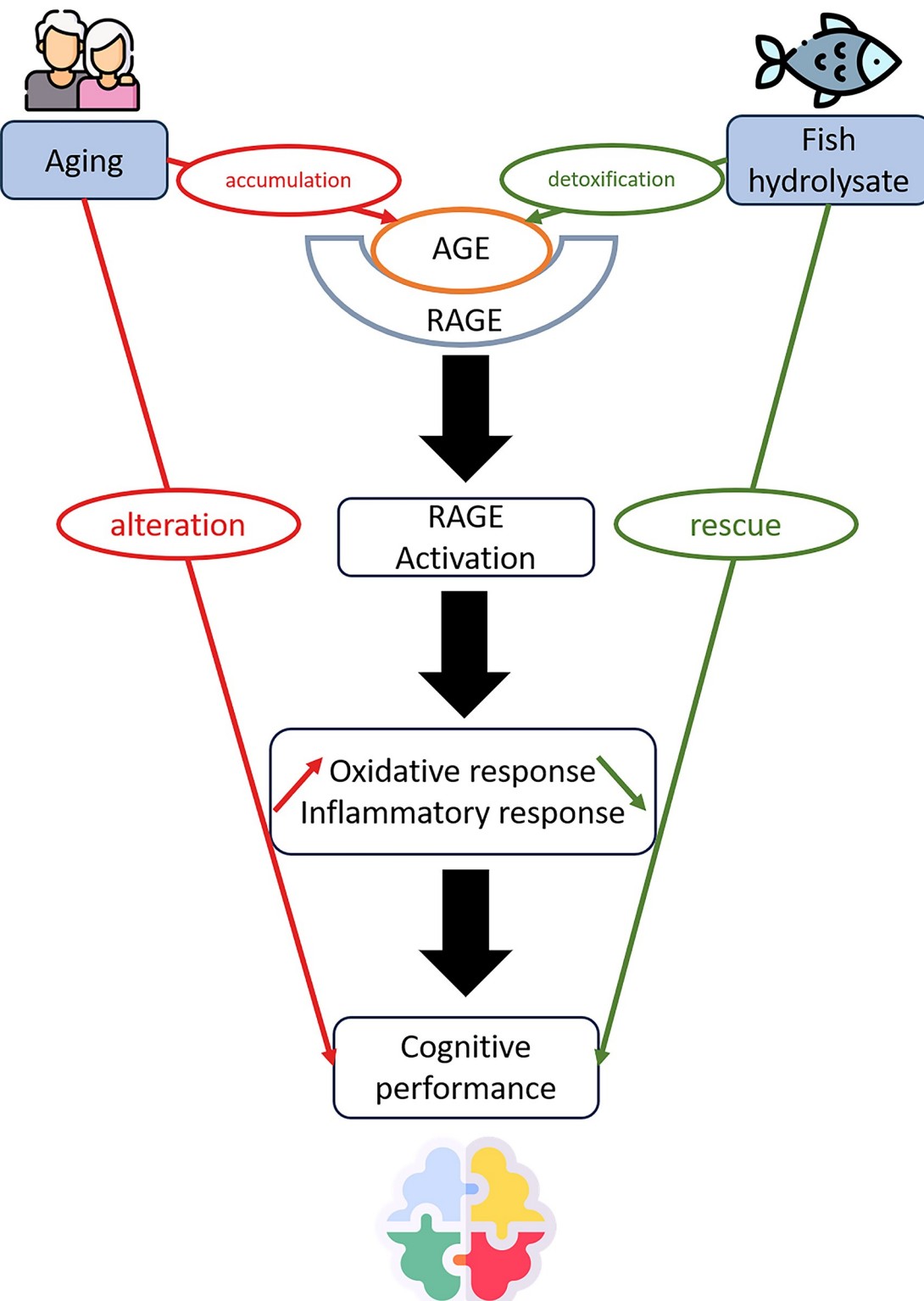

**Fig 4. Neuroprotective effects of fish hydrolysate supplementation on cognitive function, inflammatory and oxidative response through the AGE-RAGE axis.**

expression of genes involved in inflammatory and oxidative pathways. FH supplementation regulated the genetic expression of IL-6 induced by D-gal administration in mice. The AGE-R-AGE binding is a potential mechanism underlying the AGE-induced inflammatory response. Similar to toll-like receptors (TLRs), RAGEs are also considered to be pattern recognition receptors predominantly believed to be involved in the recognition of endogenous molecules released during physiological stress or chronic inflammation. RAGE is expressed in a myriad of immune cell types [67, 68]. With age, microglial cells the resident innate immune cells of the central nervous system—become chronically active, leading to sustained release of pro-inflammatory cytokines such as IL-6 and TNF-a, known as contributors to cognitive decline [50]. The elevation of these cytokines, regulated by the activation of the redox-sensitive transcription factor NF-κB release to the nucleus by AGE-RAGE interaction, can initiate a cascade of neuroinflammatory processes, including neuronal death, reduced brain volume and cortical thinning [10]. Moreover, we demonstrated that IL-6 genic expression was negatively correlated with saline and FH profiles. Therefore, one of the pathways of FH supplementation to prevent cognitive impairment was the reduction of IL-6 gene expression.

In addition to the improvements in cognitive function parameters, our study demonstrated that the FH supplementation enhanced catalase genic expression, SOD1 protein expression, following D-gal administration in mice. As in our previous study conducted in a model of aged mice, we demonstrated that FH supplementation significantly improved SOD1 expression [38]. A crucial aspect of neurodegeneration involves the declining efficiency of the antioxidant system, resulting in decreased capacity to reduce ROS. Studies have indicated that increased accumulation of AGEs leads to reduced expression of various antioxidant enzyme systems. Indeed, SOD1 and glutathione peroxidase (GPx) are diminished in the plasma of both familial and sporadic amyotrophic lateral sclerosis patients [69]. Also, SOD1 protein expression was positively correlated with recognition indices of behavioral tests and saline and FH profiles. Peripheral artery diseases have unveiled a correlation between AGEs and the total antioxidant system [70]. Thus, the increase of SOD1 protein expression observed with the FH supplementation may explain the benefits highlighted in cognitive function tests.

Finally, although this study successfully showed the beneficial effects of FH supplementation on cognitive functions through the regulation of oxidative stress and inflammation pathways via the AGE-RAGE axis, it presented some limitations. AGE-RAGE axis is not limited to these two pathways. Indeed, increased oxidative stress is a major hallmark of aging. Firstly, AGE-RAGE interaction triggers inflammation, leading to Rac1 upregulation, which activates NOX, consequently increasing ROS production. Additionally, RAGE engagement by AGEs induces a hyper-responsive state in macrophages and monocytes, enhancing the secretion of pro-inflammatory cytokines like IL-12, insulin-like growth factor, and TNF-α [71, 72], further exacerbating ROS and reactive nitrogen intermediates. It would have been intriguing to measure these parameters alongside microglial reactivity. Secondly, a potential mechanism underlying AGE-induced harm involves programmed cell death [73]. AGEs have been documented to trigger apoptosis in diverse cultured cells, such as microvascular cells, neuronal cells, fibroblasts, and renal mesangial cells [74]. It would have been interesting to assess the activity of the pro-apoptotic protein Bax, the expression of the anti-apoptotic protein Bcl-2, and the integrity of mitochondrial genes. Thirdly, mitochondrial damage induced by D-gal could contribute to neuronal decline by increasing oxidative stress. Indeed, the administration of D-gal to animals can induce aspects of brain aging similar in many ways to human brain aging, including increased mitochondrial DNA mutation and impaired mitochondrial structure [75–79]. Mitochondrial damage accumulates over time and progressively contributes to neuronal decline as one ages, much as in neurodegenerative conditions [80]. Interestingly, it has been previously demonstrated that n-3 LC-PUFAs in the inner mitochondrial membrane affect oxidative

stress, suppressing production of and scavenging ROS, with neuroprotective benefits. Consequently, our FH supplementation could also act by improving mitochondrial function that will contribute to prevent oxidative stress and to limit cognitive alteration. This point has to be further assessed by measuring mitochondrial function markers. Finally, the study was carried out on male mice. Notably, the effects of RAGE on glucose homeostasis varied based on sex. Female RAGE−/− mice fed a high-fat diet exhibited markedly enhanced glucose and insulin tolerance compared to their male counterparts [81]. To confirm the effect of FH supplementation, a new study should be conducted on female mice.

As outlined in this study, FH supplementation represented a promising approach to positively modulate inflammatory and antioxidant responses through the AGE-RAGE axis, thereby contributing to neuroprotective properties and thus prevent cognitive impairment during the aging process.

## Conclusion

The results indicated that FH had protective effect against cognitive deficits induced by D-galactose through raising the protein expression of SOD1 and decreased genetic expression of IL-6 and AGE. Furthermore, in the context of preventive and personalized nutrition, FH emerged as a promising option for preventing age-related decline in memory performance.

## Acknowledgments

The authors would like to thank Gregory Artaxet for animal care.

## Author Contributions

**Conceptualization:** Mathilde Chataigner, Corinne Joffre, Anne-Laure Dinel.

**Funding acquisition:** Véronique Pallet, Elodie Bouvret, Corinne Joffre, Anne-Laure Dinel.

**Investigation:** Camille Mougin, Céline Lucas.

**Methodology:** Camille Mougin, Mathilde Chataigner, Céline Lucas.

**Project administration:** Corinne Joffre, Anne-Laure Dinel.

**Supervision:** Véronique Pallet, Elodie Bouvret, Corinne Joffre, Anne-Laure Dinel.

**Validation:** Corinne Joffre, Anne-Laure Dinel.

**Writing – original draft:** Camille Mougin.

**Writing – review & editing:** Elodie Bouvret, Corinne Joffre, Anne-Laure Dinel.

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
