## [Decision Letter · Decision Letter 0]

1 Jul 2024

PONE-D-24-19755Dietary Marine Hydrolysate Alleviates D-Galactose-Induced Brain Aging by Attenuating Cognitive Alterations, Oxidative Stress and Inflammation through the AGE-RAGE AxisPLOS ONE

Dear Dr. DINEL,

Thank you for submitting your manuscript to PLOS ONE. After careful consideration, we feel that it has merit but does not fully meet PLOS ONE’s publication criteria as it currently stands. Therefore, we invite you to submit a revised version of the manuscript that addresses the points raised during the review process.

We look forward to receiving your revised manuscript.

Kind regards,

Ming-Chang Chiang

Academic Editor

PLOS ONE

Journal Requirements:

2. Thank you for stating the following financial disclosure: "This work is part of the Optimyss project, which has been funded by the National Agency of Research (ANR France) and Abyss Ingredients in the context of the national French project “Plan de relance” (ANR 21-PRRD-0058-01)." 

3. Thank you for stating the following in the Competing Interests section: "Céline Lucas, Véronique Pallet, Corinne Joffre and Anne-Laure Dinel report no disclosures. Abyss Ingredients funds Camille Mougin, Mathilde Chataigner, and Elodie Bouvret. Mathilde Chataigner: employee of Abyss Ingredients; Elodie Bouvret: employee of Abyss Ingredients. This work is part of a collaborative project named Optimyss which has been funded by the National Agency of Research (ANR France) and Abyss Ingredients in the context of national French project “Plan de relance”. Mathilde Chataigner and Elodie Bouvret work for Abyss Ingredients and provide the fish hydrolysate, described in the patent number FR3099339(B1) in which Mathilde Chataigner is cited as an inventor; Camile Mougin was recruited for this specific research program by NutriNeuro."

Reviewers' comments:

Reviewer's Responses to Questions

**Comments to the Author**

1. Is the manuscript technically sound, and do the data support the conclusions?

Reviewer #1: Yes

Reviewer #2: Yes

2. Has the statistical analysis been performed appropriately and rigorously? 

Reviewer #1: Yes

Reviewer #2: Yes

3. Have the authors made all data underlying the findings in their manuscript fully available?

Reviewer #1: No

Reviewer #2: Yes

4. Is the manuscript presented in an intelligible fashion and written in standard English?

Reviewer #1: Yes

Reviewer #2: Yes

5. Review Comments to the Author

Reviewer #1: This manuscript investigates the effects of a fish hydrolysate rich in low molecular weight peptides and n-3 LC-PUFAs on cognitive function, inflammation, and oxidative stress in a D-galactose induced accelerated aging mouse model. The authors demonstrate that the fish hydrolysate protects against cognitive deficits induced by D-galactose, reducing inflammation and oxidative stress via the AGE-RAGE axis.

Strengths:

- The study uses a well-established D-galactose induced aging model, which allows for the investigation of age-related cognitive decline and other physiological changes.

- The authors use well-established behavioral tests, including the Y-maze, Morris water maze, to assess spatial learning and memory.

- They investigate both gene expression and protein expression, providing a more complete picture of the molecular mechanisms involved.

- The study uses multivariate analysis to identify potential links between different pathways and the observed effects of the fish hydrolysate.

- The manuscript is well-written, with clear and concise language.

Weaknesses:

- Data download links for gene/protein expression data, behavioral test results, etc. are not specified - I couldn't find them in the manuscript text nor in the Supplementary Material. Providing links directly in the Data Availability section of the submission form would be helpful

- Sample size is on the small side, especially for the D-gal treated with control diet group (n=8)

- Correction for multiple statistical comparisons is not performed or not mentioned

- Assumptions: It is crucial to assess whether the assumptions of the statistical tests were met. For instance, for the t-tests and ANOVA, the data should be normally distributed and have equal variances. The authors should discuss these assumptions and address any potential violations.

- Specificity of Effects: The discussion merely states that the fish hydrolysate "effectively prevented cognitive impairments" without elaborating on how the peptides and n-3 LC-PUFAs might be working at the molecular level. It's crucial to explore the specific pathways these components might be targeting within the AGE-RAGE axis, as well as other potential mechanisms (e.g., mitochondrial function, neurogenesis).

- Comparison with Other Interventions: The discussion mentions the study's limitations in focusing only on males, but fails to discuss how the findings compare with other interventions for cognitive decline and aging, such as exercise, other dietary supplements, or even pharmacotherapy. This comparison would help position the findings within the broader context of research.

- Causality: While the study establishes a correlation between fish hydrolysate and improved cognitive function, it doesn't prove a causal relationship. The discussion could acknowledge this limitation and suggest further research using more controlled experimental designs or interventions that directly manipulate specific molecular pathways.

Recommendation:

Overall, this manuscript provides valuable insights into the potential benefits of fish hydrolysate in mitigating age-related cognitive decline. The study is well-designed and executed, and the results are generally convincing. I recommend the manuscript be accepted after minor revisions. The authors should address the limitations outlined above, providing additional analysis or discussion, and strengthening the conclusions of their study.

Reviewer #2: The manuscript aims to identify whether the fish hydrolysate could restore memory decline in aging mouse models. Although, this is a bit preliminary study, the data presented is very solid. My comments are as follows:

1. Author tested only male animal. They should test whether the fish hydrolysate will impact memory decline during aging in female animals.

2. Did the fish hydrolysate enhances the weight of the animal?

3. All data point should be indicated in each figure.

4. How global gene expression profile was made in Figure 3? qPCR or any other method? Please specify this information in text and figure legend.

5. Author should explain how gene expression change following fish hydrolysate could affect spatial memory. In absence of this description, it is hard to appreciate the link between change in gene expression and deficits in memory.

6. PLOS authors have the option to publish the peer review history of their article (what does this mean?). If published, this will include your full peer review and any attached files.

Reviewer #1: No

Reviewer #2: **Yes: **Sourav Banerjee

---

## [Author Response · Author response to Decision Letter 0]

26 Jul 2024

We would like to thank the reviewers for their insightful comments. Point-by-point answers to the referee’s comments are provided below. Of note, we have highlighted in yellow in the revised version the modifications according to reviewer’s comments. These changes, we think, have significantly improved the quality of the manuscript

Journal Requirements:

2. Thank you for stating the following financial disclosure: "This work is part of the Optimyss project, which has been funded by the National Agency of Research (ANR France) and Abyss Ingredients in the context of the national French project “Plan de relance” (ANR 21-PRRD-0058-01)." 

We added this comment in the cover letter.

3. Thank you for stating the following in the Competing Interests section: "Céline Lucas, Véronique Pallet, Corinne Joffre and Anne-Laure Dinel report no disclosures. Abyss Ingredients funds Camille Mougin, Mathilde Chataigner, and Elodie Bouvret. Mathilde Chataigner: employee of Abyss Ingredients; Elodie Bouvret: employee of Abyss Ingredients. This work is part of a collaborative project named Optimyss which has been funded by the National Agency of Research (ANR France) and Abyss Ingredients in the context of national French project “Plan de relance”. Mathilde Chataigner and Elodie Bouvret work for Abyss Ingredients and provide the fish hydrolysate, described in the patent number FR3099339(B1) in which Mathilde Chataigner is cited as an inventor; Camile Mougin was recruited for this specific research program by NutriNeuro."

We added this comment in the cover letter.

Reviewers' comments:

Reviewer's Responses to Questions

Comments to the Author

1. Is the manuscript technically sound, and do the data support the conclusions?

Reviewer #1: Yes

Reviewer #2: Yes

2. Has the statistical analysis been performed appropriately and rigorously? 

Reviewer #1: Yes

Reviewer #2: Yes

3. Have the authors made all data underlying the findings in their manuscript fully available?

Reviewer #1: No

Reviewer #2: Yes

4. Is the manuscript presented in an intelligible fashion and written in standard English?

Reviewer #1: Yes

Reviewer #2: Yes

5. Review Comments to the Author

Reviewer #1: This manuscript investigates the effects of a fish hydrolysate rich in low molecular weight peptides and n-3 LC-PUFAs on cognitive function, inflammation, and oxidative stress in a D-galactose induced accelerated aging mouse model. The authors demonstrate that the fish hydrolysate protects against cognitive deficits induced by D-galactose, reducing inflammation and oxidative stress via the AGE-RAGE axis.

Strengths:

- The study uses a well-established D-galactose induced aging model, which allows for the investigation of age-related cognitive decline and other physiological changes.

- The authors use well-established behavioral tests, including the Y-maze, Morris water maze, to assess spatial learning and memory.

- They investigate both gene expression and protein expression, providing a more complete picture of the molecular mechanisms involved.

- The study uses multivariate analysis to identify potential links between different pathways and the observed effects of the fish hydrolysate.

- The manuscript is well-written, with clear and concise language.

Weaknesses:

- Data download links for gene/protein expression data, behavioral test results, etc. are not specified - I couldn't find them in the manuscript text nor in the Supplementary Material. Providing links directly in the Data Availability section of the submission form would be helpful

We apologize for this mistake. The link for the data availability section has been added in the text line 194 https://doi.org/10.57745/WRKTLL

- Sample size is on the small side, especially for the D-gal treated with control diet group (n=8)

The number of animals in the D-gal group was defined according to previous publications on D-Gal model. Some studies used less animals than us (Jiangang et al, 2007) some of them used the same number (n=8) (Jun et al, 2007; Shih-Ching et al, 2003; Song et al, 1999) and some of them used more animals than us : n=9-10 (Cui et al, 2006, Jun et al, 2006; Zhang et al, 2005 and 2007) and n=11-13 (Shen et al, 2002; Wei et al, 2005). In order to fully respect the 3R (Replacement, Reduction, Refinement), in particular Reduction consisting in minimizing the number of animals used, we decided to include 9 mice and unfortunately, one died.

We included more mice in the supplemented group because there is always more variability when it involves nutrition.

- Correction for multiple statistical comparisons is not performed or not mentioned

We realized Multiple comparisons t-test (Fisher LSD), which makes no correction for multiple comparisons. As we only made a few planned comparisons (3 groups), it is recommended to set the significance level (or the meaning of the confidence interval) for each individual comparison without correction for multiple comparisons. In our case, the 5% traditional significance level applies to each individual comparisons, rather than the whole family of comparisons.

- Assumptions: It is crucial to assess whether the assumptions of the statistical tests were met. For instance, for the t-tests and ANOVA, the data should be normally distributed and have equal variances. The authors should discuss these assumptions and address any potential violations.

We checked the normality (Shapiro test) and the equality of variances for each data. All our data are normally distributed except IL1beta for FH group, p21 for D-Gal group, RAGE for FH group, SOD1 for saline group. Since the equality of variances is respected for each group and for each variable, we can use the ANOVA test because it is a test that is fairly robust to deviations from normality.

- Specificity of Effects: The discussion merely states that the fish hydrolysate "effectively prevented cognitive impairments" without elaborating on how the peptides and n-3 LC-PUFAs might be working at the molecular level. It's crucial to explore the specific pathways these components might be targeting within the AGE-RAGE axis, as well as other potential mechanisms (e.g., mitochondrial function, neurogenesis).

In this study, we principally focused on the positive impact of fish hydrolysate on cognitive impairment by evaluating the AGE-RAGE axis. As we demonstrated, by positively modulating inflammatory and antioxidant responses through the AGE-RAGE axis, the fish hydrolysate supplementation can prevent cognitive impairments. We elaborated some hypothesis and explored the specific pathways involved. Indeed, we demonstrated that fish supplementation:

- prevented the AGE-induced inflammatory response

- promoted SOD1 expression and limited AGE-induced oxidation response

But we totally agree with the reviewer considering that other potential mechanisms can be involved in the positive effect. In the discussion, we cited these two pathways and the programmed cell death pathway (line 377-387). We could also add the mitochondrial damage pathway. Indeed, the administration of D-galactose to animals can induce aspects of brain aging similar in many ways to human brain aging, including increased mitochondrial DNA mutation and impaired mitochondrial structure (Banji et al., 2014, Kumar et al., 2009, Lei et al., 2008, Prakash and Kumar, 2013, Ullah et al., 2015). Mitochondrial damage accumulates over time and progressively contributes to neuronal decline as one ages, much as in neurodegenerative conditions (Grimm et al, 2016). Interestingly, it has been previously demonstrated that n-3 LC-PUFAs in the inner mitochondrial membrane affect oxidative stress, suppressing production of and scavenging ROS, with neuroprotective benefits. Consequently, our fish hydrolysate supplementation could also act by improving mitochondrial function that will contribute to prevent oxidative stress and to limit cognitive alteration. This point has to be further assessed by measuring mitochondrial function markers. 

This point has been added in the revised manuscript line 387 to 396: “Thirdly, mitochondrial damage induced by D-gal could contribute to neuronal decline by increasing oxidative stress. Indeed, the administration of D-gal to animals can induce aspects of brain aging similar in many ways to human brain aging, including increased mitochondrial DNA mutation and impaired mitochondrial structure. Mitochondrial damage accumulates over time and progressively contributes to neuronal decline as one ages, much as in neurodegenerative condition. Interestingly, it has been previously demonstrated that n-3 LC-PUFAs in the inner mitochondrial membrane affect oxidative stress, suppressing production of and scavenging ROS, with neuroprotective benefits. Consequently, our FH supplementation could also act by improving mitochondrial function that will contribute to prevent oxidative stress and to limit cognitive alteration. This point has to be further assessed by measuring mitochondrial function markers.”

- Comparison with Other Interventions: The discussion mentions the study's limitations in focusing only on males, but fails to discuss how the findings compare with other interventions for cognitive decline and aging, such as exercise, other dietary supplements, or even pharmacotherapy. This comparison would help position the findings within the broader context of research.

We totally agree with the reviewer. Several studies have investigated the impact of nutrition or exercise on cognitive decline. However, it is difficult, in mice model, to compare the impact of those interventions to our FH supplementations since we cannot define finely the level of memory ability. Nevertheless, in human, a meta-analysis demonstrates that physical exercise (aerobic and resistance exercise of at least moderate intensity), improves cognitive function in the over 50s, regardless of the cognitive status of participants (Northey et al, 2018). Moreover, several publications from our lab demonstrate the impact of dietary supplements on cognitive decline in mice and human (Labrousse et al, 2012, Chataigner et al, 2020, Bensalem et al, 2018, Dumetz et al, 2020). Our results confirmed the crucial impact of preventive and personalized nutrition in healthy ageing. 

A paragraph has been added : “Our results can be compared with other interventional studies questioning cognitive decline. In human, a meta-analysis demonstrates that physical exercise (aerobic and resistance exercise of at least moderate intensity), improves cognitive function in the over 50s, regardless of the cognitive status of participants (Northey et al, 2018). Moreover, several publications from our lab demonstrate the impact of dietary supplements on cognitive decline in mice and human (Labrousse et al, 2012, Chataigner et al, 2020, Bensalem et al, 2018, Dumetz et al, 2020).” in the final manuscript to position our findings within the context of research in line 325-330.

- Causality: While the study establishes a correlation between fish hydrolysate and improved cognitive function, it doesn't prove a causal relationship. The discussion could acknowledge this limitation and suggest further research using more controlled experimental designs or interventions that directly manipulate specific molecular pathways.

We totally agree with the reviewer. However, it is difficult to establish a causal relationship between nutritional supplementation (composed of 3 different types of nutrients) and cognitive performances. Most of the studies on nutrition and aging failed to report such causality link (but report correlation link). In the lab, we already showed a correlation link between nutrition and cognition (Labrousse et al, 2012, Chataigner et al, 2021, Chataigner et al, 2021, Chataigner et al, 2020, Bensalem et al, 2018, Dumetz et al, 2020). The best way to explore a causality link is to work with KO mice but in our case, it seems difficult to generate KO mice for the receptors involved in the action of low molecular weight peptides and of omega-3. 

Recommendation:

Overall, this manuscript provides valuable insights into the potential benefits of fish hydrolysate in mitigating age-related cognitive decline. The study is well-designed and executed, and the results are generally convincing. I recommend the manuscript be accepted after minor revisions. The authors should address the limitations outlined above, providing additional analysis or discussion, and strengthening the conclusions of their study.

Reviewer #2: The manuscript aims to identify whether the fish hydrolysate could restore memory decline in aging mouse models. Although, this is a bit preliminary study, the data presented is very solid. My comments are as follows:

1. Author tested only male animal. They should test whether the fish hydrolysate will impact memory decline during aging in female animals.

We totally agree with Reviewer 2, we have to test the effect of the fish hydrolysate in female animals. This is the next step. We already mentioned this point in the paragraph concerning the limitations of the study in the Discussion section. We began with male animals to first evaluate the effect of the fish hydrolysate on a reduced cohort, in agreement with the 3R (Replacemen

---

## [Editor Report · Decision Letter 1]

14 Aug 2024

Dietary Marine Hydrolysate Alleviates D-Galactose-Induced Brain Aging by Attenuating Cognitive Alterations, Oxidative Stress and Inflammation through the AGE-RAGE Axis

PONE-D-24-19755R1

Dear Dr. DINEL,

We’re pleased to inform you that your manuscript has been judged scientifically suitable for publication and will be formally accepted for publication once it meets all outstanding technical requirements.

Kind regards,

Ming-Chang Chiang

Academic Editor

PLOS ONE

Additional Editor Comments (optional):

**The manuscript has been greatly improved and completed with new experiments that have added quality and clarity to the message. The manuscript is acceptable for publication.**
---

## [Editor Report · Acceptance letter]

19 Aug 2024

PONE-D-24-19755R1 

PLOS ONE

Dear Dr. DINEL, 

I'm pleased to inform you that your manuscript has been deemed suitable for publication in PLOS ONE. Congratulations! Your manuscript is now being handed over to our production team.

Kind regards, 

on behalf of

Dr. Ming-Chang Chiang 

Academic Editor

PLOS ONE